# Analyzing enterprise asset structure and profitability using cloud computing and strategic management accounting

Wenquan Shi [1,2] *

1 Faculty of Economics and Management, Suzhou Polytechnic Institute of Agriculture, Suzhou City, China,
2 University of Cordilleras, Baguio City, Benguet, Philippines

* wqshi@szai.edu.cn

## Abstract

The study expects to further exploring the role of asset structure in enterprise profitability, and analyze the relationship between them in detail. Taking the express industry as the research object, from strategic management accounting, the study uses edge computing and related analysis tools and compares the financial and non-financial indicators of existing express enterprises. The study also discusses the differences between asset structure allocation and sustainable profitability, and constructs the corresponding analysis framework. The results reveal that SF's total assets are obviously large and the profit margin increases. While the total assets of other express enterprises are small, and the express revenue drops sharply. Heavy assets can improve the enterprises' profitability to a certain extent. SF has a good asset management ability. With the support of the capital market, SF's net asset growth ability has been greatly improved. The edge computing method used has higher local data processing ability, and the analysis framework has higher performance than the big data processing method. The study can provide some research ideas and practical value for the asset structure analysis and profitability evaluation of express enterprises.

**Data Availability Statement:** All relevant data are within the manuscript and its Supporting Information files.

**Funding:** The author(s) received no specific funding for this work.

## Introduction

With the rapid development of the economy and society, people's demands continue to increase, which has prompted the rapid development of the e-commerce industry [1]. According to the National Bureau of Statistics, China's national e-commerce transaction volume has reached 34.18 trillion Chinese yuan; moreover, online retail sales have reached 10.63 trillion Chinese yuan, a year-on-year increase of 16.5% [2]. E-commerce has promoted the development of the express delivery industry. From 2013 to 2019, the total amount of express delivery business in China has proliferated. Although the growth rate has declined in the past two years, it has maintained more than 20% [3]. In 2019, express service enterprises' business volume reached 50.71 billion pieces, with a year-on-year increase of 26.6% [4]. The current express delivery industry in China is dominated by five major enterprises, including SF

**Competing interests:** The authors have declared that no competing interests exist.

Express and other self-operated express enterprises. With the transformation and upgrading of China's economy, the differential development of the express delivery industry in different regions needs to be further improved, promoting enterprises' production factors and regional differential consumption [5]. Among the express enterprises in China, SF Express takes the high-end route. It can provide customers with safe, fast, and high-quality logistics and transportation services, while other small and medium express enterprises use low-cost and high-circulation models in e-commerce express delivery. Occupying an important position, the express service enterprise headed by JD's self-operated express service also has considerable revenue [6]. As the dividend growth rate of the express delivery industry has slowed down, a new normal has emerged in the industry's competition. Among them, SF Express's asset-heavy management structure faces cost control and asset utilization issues based on enjoying user dividends. The representative asset-light operation method, based on enjoying user stickiness, also faces price war and enterprise development [7]. Due to the significant differences in the enterprise's business methods and profit sources, studying different enterprise asset structures and profitability is significant to relevant enterprises' sustainable development.

At present, research on enterprise asset structure mostly adopts empirical methods. Different enterprises and industries have different asset structures. Although there are many studies from the perspective of enterprise profitability, the current research lacks long-term development considerations for the enterprise [8]. Moreover, the data emerging from the express delivery industry cannot be effectively used, which severely restricts the development of the express delivery industry [9]. Cloud computing is a distributed computing architecture. Data and services from the network's central node to the edge node of the network logic can effectively process local data and reduce central server dependence by moving the computing of applications. This feature can effectively solve complicated data processing in the express delivery industry [10]. Strategic management accounting analyzes industry competitors' characteristics based on financial indicators, which considers the short-term and long-term effects and focuses on the enterprises' future development and sustainable profitability. It can provide new research ideas for evaluating the rationality of enterprise asset structure configuration [11]. Therefore, with enterprise asset structure and profitability as the research purposes, analyzing the differences between different express delivery enterprises from cloud computing and strategic management accounting can provide development proposals for relevant enterprises. The results can provide relevant research ideas for the in-depth understanding of enterprise asset structure and profitability analysis and promote the express delivery industry's sustainable and healthy development.

Related previous literature is first combed here. Then, combined with the reality of the express industry, edge computing is adopted to analyze the relationship between the asset structure and profitability of SF Express to find the advantages and disadvantages of the heavy asset structure of SF Express. In addition, SF Express is compared with STO. Express, YTO Express, ZTO Express, BEST Express, and YUNDA Express. The difference in the sustainable profitability of express enterprises under the light asset structure and heavy asset structure is discussed through the analysis of financial and non-financial indicators. Furthermore, from the perspective of strategic management accounting, the problems faced by the existing domestic express delivery industry are analyzed by building the edge computing data processing model, concentrating on the relationship between corporate asset structure and profitability. The research result has certain guiding value for the development of the express delivery industry and other related industries. The innovation of this research is as follows. (1) The introduction and application of cloud computing to analyze the data of the express delivery industry can effectively improve the data processing and analysis capabilities of relevant enterprises. (2) According to strategic management accounting theory, the short-term and long-

term development of express delivery enterprises can focus on future development and sustainable profitability. (3) Based on theoretical research, specific case data can analyze and study the asset structure and profitability of different express delivery enterprises. The total time taken for the study is 6 month. The cost related to the study during this period is 40,000 RMB including data collection and server computing.

## Literature review

### Research progress of enterprise asset structure

Asset structure refers to the proportion of various assets in an enterprise's total assets, essentially referring to the proportion of fixed investment, securities investment, and liquidity investment. Many scholars have researched the asset structure. Yang et al. (2017) pointed out that market interest rates and national policies will affect bank investment's asset structure. They found that different interest rate policies significantly impacted the real estate industry's proportion [12]. Paniagua et al. (2018) used multivariate analysis to estimate the connection between enterprise governance ownership structure and enterprise financial performance. They found that the complementary linear and nonlinear multivariate regression analysis could effectively enhance enterprise asset structure analysis [13]. Nguyen et al. (2019) determined the capital structure and adjustment mechanism determinants that determined the target leverage. The fixed-effect model estimation found that factors, such as growth opportunities, enterprise-scale, tangible fixed assets, and the enterprise's unique characteristics, would positively affect the enterprises' target capital structure [14]. Dhodary (2019) used descriptive and causal comparative research design to test the determinants of capital structure. The results reflected that asset tangibility, profitability, liquidity, and interest coverage were the main determinants of the capital structure of trade and manufacturing enterprises in Nepal [15]. Sumani et al. (2020) tested the impact of capital structure and liquidity policy on enterprise governance and found that enterprise governance had a significant positive impact on capital structure. However, enterprise governance had a significant adverse effect on liquidity policy, and enterprise governance negatively impacted the capital structure [16]. Legenchuk et al. (2020) believed that securitization could diversify financing sources, effectively manage the balance sheet structure of enterprises, and significantly improve the liquidity level of their assets [17].

### Research progress of enterprise profitability

Profitability refers to the enterprise's ability to obtain profit, which is usually expressed as the amount and level of enterprise income in a period. The profitability indicator includes six items: operating profit rate, cost and expense profit rate, surplus cash protection multiple, return on total assets, return on net assets, and capital return. There are many reports on the profitability of the enterprise. Dang et al. (2019) studied the impact of Vietnam's economic growth, enterprise size, capital structure, and profitability on enterprise value. The results showed that enterprise size and profitability were positively correlated with enterprise value, capital structure was negatively correlated with enterprise value, and economic growth had no impact on enterprise value [18]. Podile et al. (2019) held that financial soundness could be measured by operating efficiency, while the operating efficiency of an enterprise was determined by the profits earned by the enterprise, and the profitability of the enterprise was analyzed through various profit margins and profitability ratios related to investment [19]. Li (2021) pointed out that the enterprise development aimed to obtain profits, and profitability was the core index to measure the development status and prospect of enterprises [20]. Otekunrin et al. (2021) thought that the payback period of trade receivables was negatively

correlated with profitability, the payment period of trade receivables was positively correlated with profitability, and the cash turnover cycle was positively correlated with profitability. Obviously, working capital management and profitability were related, and efficient and effective decision-making could maximize the enterprises' profitability [21].

## Association between enterprise asset structure and profitability

In China, the current investigations on the relationship between asset structure and sustainable profitability are empirical analysis and research on a particular industry's financial data. Glova et al. (2018) empirically studied the proportion of fixed assets in the total assets of the automobile manufacturing industry's listed enterprises and the research and development investment. They put forward relevant suggestions for optimizing the asset structure for the automobile manufacturing industry [22]. Mahrani and Soewarno (2018) analyzed the impact of enterprise asset allocation structure and enterprise profitability by analyzing the listed enterprises' financial statements. They found that enterprise governance and enterprise social responsibility mechanisms positively impacted financial performance and enterprise social responsibility [23]. Besides, many scholars have combined capital structure, asset structure, and profitability for empirical research. Yanikkaya et al. (2018) conducted an empirical analysis on the financial data of listed rare earth enterprises and found that both the asset structure and capital structure in the rare earth industry had a significant impact on profitability. Regarding the generally unreasonable asset structures, they proposed suggestions for optimizing the asset structure [24]. Nastiti et al. (2019) empirically analyzed the connection between high-tech listed enterprise asset structure and profitability. Finally, they found that the cash asset ratio and fixed asset ratio significantly impacted the enterprise's sustainable profitability. In contrast, the proportion of intangible assets and the return on total assets did not show any correlations [25]. Dubey et al. (2020) studied the association between enterprise asset structure and profitability based on the contingency theory and believed that the asset structure and profitability were linearly correlated [26].

## A summary of previous works

In summary, domestic experts and scholars have gained abundant research results and experience on the relationship between capital structure and profitability. However, even if the relevant scholars have conducted multi-angle and multi-level research, there is still space for improvement in the innovation and depth of the research. In addition, both foreign and domestic studies have the following common problems in terms of capital structure and profitability. a) The sample selection is not well considered. The selected samples are limited to two kinds, one is to study a certain industry, the other is to study the national listed companies. The conclusions drawn by the former are often only for the selected industries, which can only obtain finite achievements. The sample selected by the latter is slightly broad, but the conclusion is usually not of practical significance. b) The selection of indicators is relatively monotonous. Most scholars only choose two most representative indicators, ignoring the impact of other indicators, and cannot comprehensively analyze the relationship between capital structure and profitability. Consequently, the research results seem not rigorous and scientific enough. c) There is no unified model and method. Diverse methods have been adopted, including multiple regression analysis, analytic hierarchy process, and least square method, yet whether these methods are effective and direct in confirming the relationship between the two remains to be verified. Moreover, it is difficult to determine which model is more appropriate for demonstration since numerous models emerge endlessly. Besides, the asset structure of enterprises will affect their profitability, but there are few studies on the correlation between

them. Research on corporate profitability mainly focuses on influencing factors. On the contrary, few studies analyze corporate profitability from the perspective of asset structure. The research on the relationship between asset structure and profitability has not yet put forward specific mechanisms and needs further analysis. At present, the data processing of the express industry has encountered great problems, and there is no clear analysis model to describe the evaluation of enterprise assets and profitability. Therefore, mobile edge computing is introduced to alleviate the pressure of the central server. In addition, the specific modeling ideas and methods are illustrated by the strategic management theory, and the data processing and asset evaluation model is built ultimately. This research is of crucial value for promoting the healthy and sustainable development of the express delivery industry.

## Research methodology

### Edge computing enterprise model

At present, the express delivery industry needs to connect different branches, aviation enterprises, and employee data, and relevant enterprises and merchants. Because of its low system configuration and high-speed computing capabilities, many enterprises have welcomed cloud computing. Strategic management accounting enables decision-makers to grasp various internal information associated with their strategies. It is also conducive for enterprises to analyze customer information and competitors and other strategic-associated external information from a strategic perspective [27]. Cloud computing effectively solves the difficult data problem faced by the express delivery industry, while the strategic management accounting method overcomes the shortcomings of traditional accounting activities. Therefore, it has been redesigned under the traditional enterprise analysis model. Fig 1 shows the enterprise analysis model based on cloud computing and strategic management accounting theory, divided into three dimensions: data processing, asset structure analysis, and profitability analysis. The data processing is based on cloud computing, the analysis of the assert structure is divided into several enterprise management indicators, and the profitability is divided into two sections: financial indicator and non-financial indicator. A suitable analysis model is constructed based on the input data, and the final enterprise analysis results can be obtained.

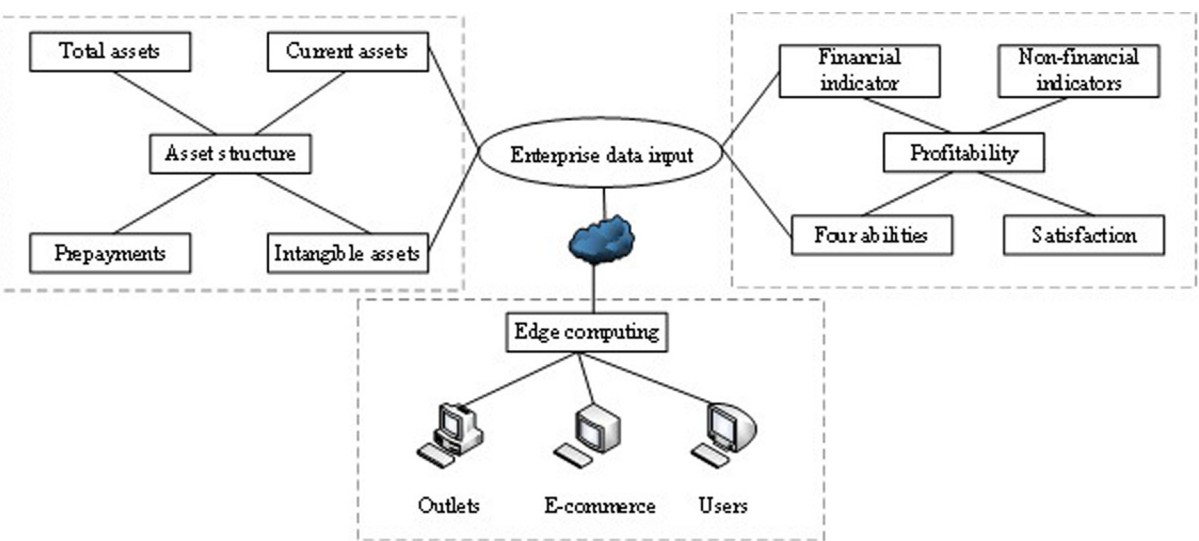

**Fig 1. The enterprise analysis model based on cloud computing and strategic management accounting.**

As shown in Fig 1, the model is divided into three parts: data preprocessing, asset structure analysis, and profitability analysis. The task types of edge computing include 5G fragmentation technology, cloud computing task migration, video surveillance, environmental and equipment monitoring, intelligent analysis, and decision support computing. The node application devices of edge computing can effectively isolate the operating environment and service environment of different users. Edge computing equipment can generate and consume data, containing functions such as data communication, data calculation, data collection, and data processing. The data is collected into the edge computing device through the communication module, to meet the requirements of multi-source data processing, and the edge computing device completes the cloud edge data cooperative operation by synchronization. The analysis of asset structure through the data of edge computing combined with the financial and non-financial data of enterprises mainly contains the following steps. First, the enterprise scale is roughly determined according to the total amount of assets. Second, the industry characteristics, industry status, management ability, and future development prospects of the enterprise is estimated approximately through the fixed assets and its composition. Third, attention should also be paid to the current assets and their structure, the proportion of various types of current assets, and the liquidity of enterprises. Finally, the suspicious item with large payment is detected according to the proportion of each item of assets in total assets, such as other receivables, deferred assets, amortized expenses, and prepaid accounts. A preliminary analysis can be carried out to investigate the correlation between the amount of above assets and the borrower's operating characteristics to further judge whether the business operation is normal. The assets can represent the strength of enterprises, which can be used by enterprises in a certain period, bring economic benefits to enterprises, or offset liabilities. Moreover, based on edge computing data, corporate profitability can be studied from the general analysis of corporate profitability and post-tax profit analysis of stock companies. The indicators reflecting the profitability of enterprises mainly include sales profit rate, cost profit rate, total asset profit rate, capital profit rate, and shareholders' equity profit rate. Ultimately, the method of joint strategic management accounting is used for classification analysis in the analysis of financial indicators.

## Edge computing data processing

The edge computing platform provides a computing model that can access the internet anytime, anywhere. Computing facilities, storage devices, and applications can be shared according to needs and ease of use. This platform can also handle problems in smart logistics centers through infrastructure, platform as a service, and software as a service. The specific processes are introduced as follows. The infrastructure provides general basic services for third parties, encapsulates service capabilities, opens service applications such as big data analysis to third parties, and directly invokes open interface services by standardizing web services. The platform provides customizable service middleware, such as data processing engine and access control, for various logistics applications through rapid development, application diversification, and custom expansion. The software provides users with required computing and storage resources through virtualization technologies, such as data storage, computing services, and resource pools, realizing resource allocation. The gateway connectsto a network facility or access point. The bottom edge device transmits data to the cloud platform via the gateway. The edge architecture comprises edge servers and edge clouds and is usually responsible for collecting data from the bottom layer, analyzing and responding to the data within one second. Edge devices are closely deployed near the edge architecture of the intelligent logistics center. Hence, logistics decisions that only require local information can be determined very quickly.

Sensing devices are designed to collect data from edge devices. Automatic navigation vehicles combine computer vision, image processing, automatic navigation, and other technologies with the system's scheduling tasks and the inventory calculation to replenish the warehouse, planning the production of the entire express center [28].

The mobile edge is calculated as follows. The Prewitt edge detection operator processes enterprise data, and two directed operators are used, each of which approximates a partial derivative:

$$p_v = \begin{pmatrix} -1 & -1 & -1 \\ 0 & 0 & 0 \\ 1 & 1 & 1 \end{pmatrix} \tag{1}$$

$$p_h = \begin{pmatrix} 1 & 0 & -1 \\ 1 & 0 & -1 \\ 1 & 0 & -1 \end{pmatrix} \tag{2}$$

In (1) and (2), $p_v$ refers to the horizontal operator, and $p_h$ refers to the vertical operator. First, the horizontal operator and the vertical operator are adopted to convolve the data to obtain two matrices. Then, the two numbers corresponding to the positions of M1 and M2 are squared and added to obtain a new matrix G, where G represents the gradient value of each data in M. Finally, the cloud computing results can be obtained through threshold processing:

$$M_{x,y} = \alpha x + \beta y + \gamma \tag{3}$$

In (3), $M_{x,y}$ indicates the edge calculation result after threshold processing, and $\alpha$ and $\beta$ are gradient coefficients. The current data value is:

$$\begin{pmatrix} -\alpha - \beta + \gamma & -\alpha + \gamma & -\alpha + \beta + \gamma \\ -\beta + \gamma & \gamma & \beta + \gamma \\ \alpha - \beta + \gamma & \alpha + \gamma & \alpha + \beta + \gamma \end{pmatrix} \tag{4}$$

The vertical and horizontal operators are defined as follows:

$$\begin{pmatrix} -a & -b & -a \\ 0 & 0 & 0 \\ a & b & a \end{pmatrix} \tag{5}$$

$$\begin{pmatrix} -a & 0 & a \\ -b & 0 & b \\ -a & 0 & a \end{pmatrix} \tag{6}$$

Two templates are adopted to convolve the current data, and the obtained directional derivative is:

$$\begin{aligned} g_x &= 2\beta(2a + b) \\ g_y &= 2\alpha(2a + b) \end{aligned} \tag{7}$$

The size of the current data gradient is:

$$G = 2(2a + b)\sqrt{\alpha^2 + \beta^2} \qquad (8)$$

Obviously:

$$2(2a + b) = 1 \qquad (9)$$

In (9), *a* refers to the horizontal coefficient, *b* denotes the vertical coefficient, and *G* indicates the gradient size. The final values of *a* and *b* are the results of data processing.

## Research data sources

Strategic management accounting is the accounting branch of enterprises' strategic management. It aims to assist senior leaders in formulating competitive strategies and implementing strategic planning, in an effort to promote a virtuous circle and continuous development of the enterprises. It analyzes issues strategically and provides customers and competitors with strategic relevance. Meanwhile, it provides internal information on the enterprises' strategies. Strategic management accounting overcomes the shortcomings of traditional management accounting, providing a large amount of non-financial information, such as quality, demand, and market share. This creates favorable conditions for enterprises to gain insight into opportunities, improve business and competitiveness, and maintain and develop long-term competitive advantages. Consequently, the needs of enterprise strategic management and decision-making can be met; more importantly, the single measurement mode of traditional accounting can be shifted. Traditional indicators to evaluate management accounting performance only emphasize "results" rather than "processes," such as return on investment, ignoring the role of relative competition indicators in performance evaluation. Strategic performance evaluation combines indicators with the strategies implemented by the enterprises and adopts different indicators according to different strategies. Moreover, the performance evaluation of strategic management accounting runs through every step of the strategic management application process, emphasizing that performance evaluation must meet the information needs of managers. Enterprises' strategic accounting management is analyzed from the value chain, competitors, operating cost management, early warning, total quality management, and comprehensive performance evaluation; it clarifies the problems in the express delivery industry and puts forward the improvements to be made by comparing various express delivery enterprises.

Strategic management accounting is classified into asset structure analysis and profitability analysis in the present work. Regarding the asset structure, the relationships among the express industry enterprise strategies, the light or heavy asset structure, and the sustainable profitability are discussed. Assets are the resources owned and controlled by the enterprises, providing a fundamental influence on the competitiveness and sustainable profitability of enterprises. Meanwhile, the express delivery industry has relatively prominent economies of scale. The increase in business volume will reduce its long-term average cost. It is also an industry with a huge investment in the early stage. Both the construction of the transshipment center and the operation of the fleet require a lot of capital. The investment of assets is the key to the development of express delivery enterprises. Regarding profitability, the financial indicators of the enterprises are analyzed. The profitable income items of different enterprises are compared and combined with enterprises' asset structures to clarify the relationship between enterprises' assets and sustainable profitability.

Indicators such as total assets, current assets, intangible assets, accounts payable are selected to analyze enterprise asset structure [29]. Total assets refer to all assets owned or controlled by

an enterprise, including current assets, long-term investments, fixed assets, intangible and deferred assets, other long-term assets, and deferred taxes, which are the total assets of the enterprise's balance sheet. Current assets refer to assets that can be realized or used by an enterprise in a business cycle of one year or more than one year, which are an essential component of enterprise assets. Intangible assets refer to non-monetary assets that are owned by a specific entity, have no specific type, and have no entity. However, they can constitute a competitive advantage or play a role in production and operation. Intellectual property rights, franchise rights, brands, human resources, customer information, and enterprise culture can all be regarded as intangible assets. Accounts payable refers to the debts incurred due to the purchase of materials, commodities, or the acceptance of labor services. These are the debt incurred by buyers and sellers due to the inconsistency of the time between the purchase and sale of materials and payment of goods.

Solvency, operating capacity, profitability, and growth capability are selected for profitability analysis. Solvency refers to the ability of an enterprise to use its assets to repay long-term and short-term debt [30]. Whether an enterprise has the ability to pay cash and repay debts is the key to its healthy survival and development. Enterprise solvency is a vital indicator that reflects an enterprise's financial status and operating capabilities. Solvency is the affordability or degree of assurance of an enterprise to repay its due debts, including the ability to repay short-term debts and long-term debts. Operating capacity refers to the efficiency and effectiveness of enterprises operating assets. The efficiency of operating assets refers to the turnover rate or turnover speed of assets. The effectiveness of business assets usually refers to the ratio between the output of the enterprise and the number of assets occupied. Profitability refers to the ability of an enterprise to obtain profits, also known as the ability of an enterprise to increase capital or property. It is often expressed as the amount and level of the enterprise's income in some time. Profitability indicators include operating profit rate, cost and expense profit rate, surplus cash protection multiple, return on total assets, return on net assets, and return on capital. Growth capability analyzes an enterprise's ability to expand its operations. It examines the enterprise's ability to expand its operations by increasing its annual income or obtaining funds using other financing methods. Growth capability refers to the future development trend and development speed of an enterprise, including the expansion of its scale, the increase of profits, and owner's equity. Enterprise growth capability refers to the ability of an enterprise to increase its asset scale, profitability continuously, and market share as the market environment changes, reflecting the future development prospects of an enterprise. Specific indicators are referred to literature [30].

## Data source and experimental environment

**(1) Data source.** All data come from cninfo (http://www.cninfo.com.cn/new/index), published annual reports of enterprises included, and the results of securities analysis from online searches. Chinese enterprises included for comparison are SF Express, STO Express, YTO Express, YUNDA Express, ZJS Express, China Postal Express and Logistics (EMS), TTK Express, and ZTO Express. International delivery express enterprises included for comparison are UPS and FedEx.

**(2) Experimental environment of cloud computing.** The simulation run of the model is conducted on a computer with Intel Core i7-3770 CPU and 16 GB memory using MATLAB 2018b. The simulation parameters are as follows: (1) the express delivery area is set to $200m \times 180m$; (2) the number of gateways, edge clouds, edge servers, and automatic navigation vehicles is set to 3, 15, 70 and 500, respectively; (3) the crossover rate is 0.05, the mutation rate is 0.01, the number of iterations is 500, and the penalty cost is 105. The information is shown in Table 1.

**Table 1. Experimental environment and specific parameters of cloud computing.**

| Parameter | Hardware environment | System parameter | Concrete value |
|---|---|---|---|
| System | Win 10 | Delivery area | 200m x180m |
| CPU | i7-3770 | Number of edge clouds | 15 |
| GPU | GeForce GTX 960 | Number of edge servers | 70 |
| Hard disk capacity | 16GB+500GB | Number of autonomous vehicles | 500 |
| Software environment | Chrome 86.0 | Crossover rate | 0.05 |
| | Matlab 2018b | Mutation rate | 0.01 |

(3) **Data processing of cloud computing.** Preprocessing primarily includes data cleaning, data integration, data reduction, and data conversion, which can greatly improve the overall quality of big data and reflect the quality of big data process. Data cleaning technology involves data inconsistency detection, noise data identification, data filtering, and data correction, which is helpful to improve the consistency, accuracy, authenticity, and availability of big data. Data integration is to integrate data from multiple data sources to form a centralized, unified database, data cube, etc., which is conducive to ameliorating the integrity, consistency, security, and availability of big data. Data reduction is to reduce the size of the data set without compromising the accuracy of the analysis results and simplify the data, including dimension reduction, data reduction, and data sampling. This process can facilitate the promotion of the value density of big data, that is, to improve the value of big data storage. Data conversion processing includes rule-based or metadata-based conversion, and model-based and learning-based conversion, to achieve data unification through conversion, which conduces to the improvement of consistency and availability of big data.

## Research results

### Analysis of asset structure

As shown in Fig 2, SF Express is taken as an example to analyze the enterprise asset structure. Over time, SF Express's monetary funds have increased from 10.43% in 2015 to 30.04%. The proportion of its accounts receivable has been maintained at about 10%, the proportion of its liquid assets has decreased annually, the proportion of its fixed assets has first increased and

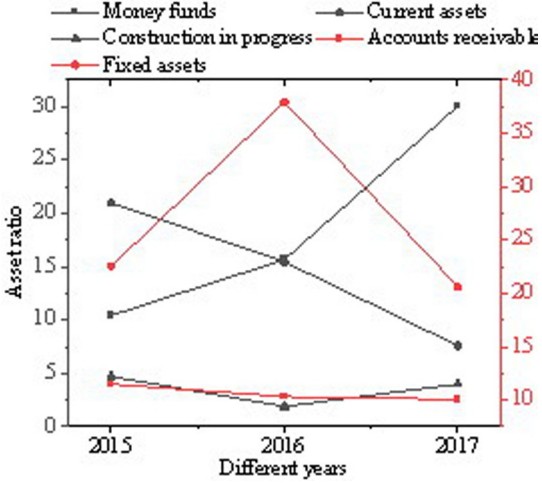

**Fig 2. Major asset accounting results of SF Express from 2015 to 2017.**

then dropped, and the proportion of its construction in progress has remained at about 4%. The above data show that SF's internal policies have not changed much; in contrast, significant changes have occurred in management strategies.

As shown in Fig 3, the enterprise asset structures of SF (Fig 3A), YTO (Fig 3B), STO (Fig 3D), and YUNDA (Fig 3C) are analyzed. SF has the largest total assets. The sum of all other enterprises is not as much as SF's total assets. Among them, current assets account for 45.03%, 53.22%, 33.70%, and 59.91%. This result shows that STO, YUNDA, and SF all focus on liquid assets. In terms of liquidity, YUNDA is the strongest. The fixed assets of SF account for 20.45%, those of STO account for 14.63%, and those of YTO and YUNDA account for 14.25% and 21.26%, respectively. YUNDA and SF have invested more in fixed assets, and none of the four enterprises have involved financial assets. In summary, the above results suggest that Chinese express enterprises' current asset structure has a large internal proportion difference, with no clear rules. SF and other enterprises' biggest difference is that the total assets are large and heavier.

As shown in Fig 4, the fixed assets, construction in progress, and intangible assets are summarized to observe the difference between SF and other enterprises, and the results are shown in Fig 4A. SF is overwhelmingly prominent in terms of heavy assets. SF's heavy assets accounted for twice those of ZTO. The intangible assets are also compared. Results suggest that SF's proportion of intangible assets is also quite different from other enterprises due to SF's unique self-operated model, making all the branches, aviation enterprises, and vehicles of SF form a mature ecological chain. Moreover, many funds have been invested in scientific research and development, which directly promotes the transformation of the enterprise asset structure.

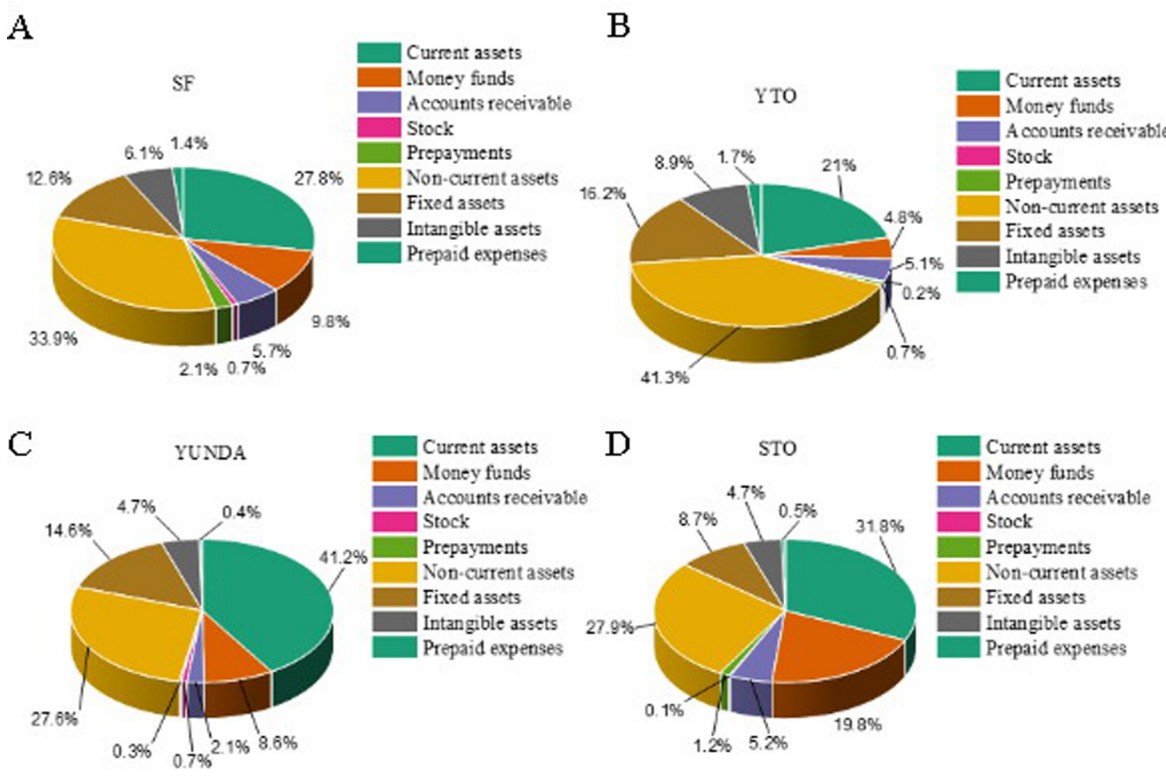

**Fig 3. Asset structures of different express enterprises.**

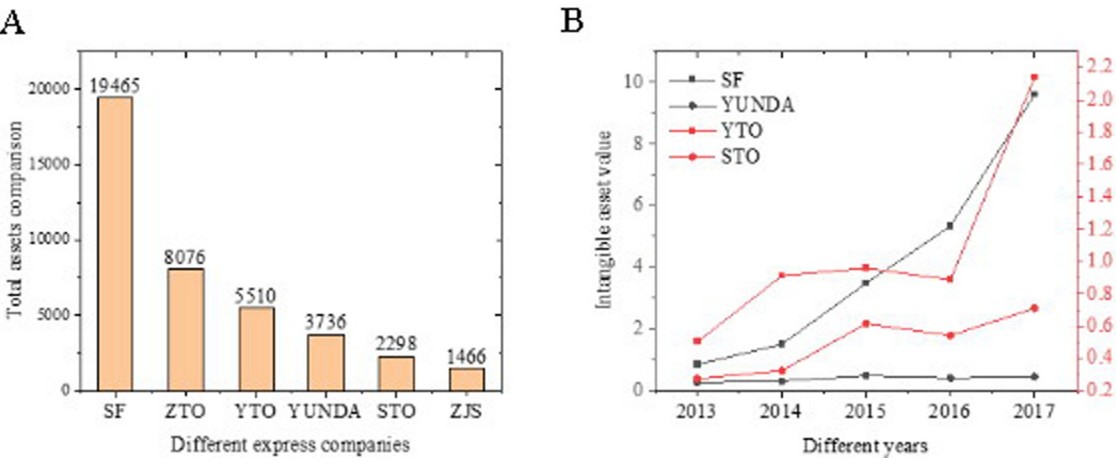

**Fig 4. Comparison of the value structure of heavy assets and intangible assets.**

## Analysis of profitability

As shown in Fig 5, SF's enterprise profitability is analyzed from four dimensions: debt repayment (Fig 5A), operation (Fig 5B), profitability (Fig 5C), and growth capability (Fig 5D). Regardless of the current ratio or the quick-freezing ratio, SF has undergone significant changes before and after the listing. The number of current assets of the enterprise is greater than the current liabilities. SF follows a relatively conservative cash management. Since 2016, SF has deliberately increased investment in infrastructure construction of aircraft and venues. The debt-to-asset ratio has almost doubled in the past two years, indicating that SF has greater confidence in borrowing business after listing, which has made full use of its operating leverage. Besides, SF's total asset turnover rate, fixed asset turnover rate, and revenue turnover rate have fluctuated dramatically in 2016 because of the significant capital investment, indicating that the enterprise asset utilization rate after the listing has increased. The profitability of SF tends to increase first and then decrease. After 2016, SF's revenue capacity has doubled, and its cost-expense ratio has also seen a substantial increase, which shows that the increase in the input unit cost-effectiveness of SF has doubled. In terms of growth capacity, the growth rate of SF's net assets per share has maintained more than 40% in the last two years. Its growth capacity has been dramatically improved after receiving the support of the capital market.

## Comparison and analysis of enterprises

Fig 6A displays the horizontal comparison results of Chinese express enterprises' debt solvency from the four dimensions of debt repayment (Fig 6A), operation (Fig 6B), profitability (Fig 6C), and growth capability (Fig 6D). Compared with other enterprises, SF's debt repayment ability is not high, and its borrowing is increasing, which is also a possible risk of heavy asset investment. In terms of the enterprise's operational capabilities, all express enterprises' total asset turnover rate is maintained at around 1. There are more monthly accounts of SF in accounts receivable, which is insufficient compared to other enterprises. Among them, the turnover rate of YTO is 60%, and SF is 24 times higher than YUNDA. This result also shows that SF's total assets are paid to other enterprises but are not settled in time. The collection period of the enterprise cooperating with SF is relatively long, which undoubtedly increases other enterprises' financial risk. Although the absolute amount of SF's overall revenue and net profit is the highest in the industry, its profit percentage is weak. Except for the sales gross

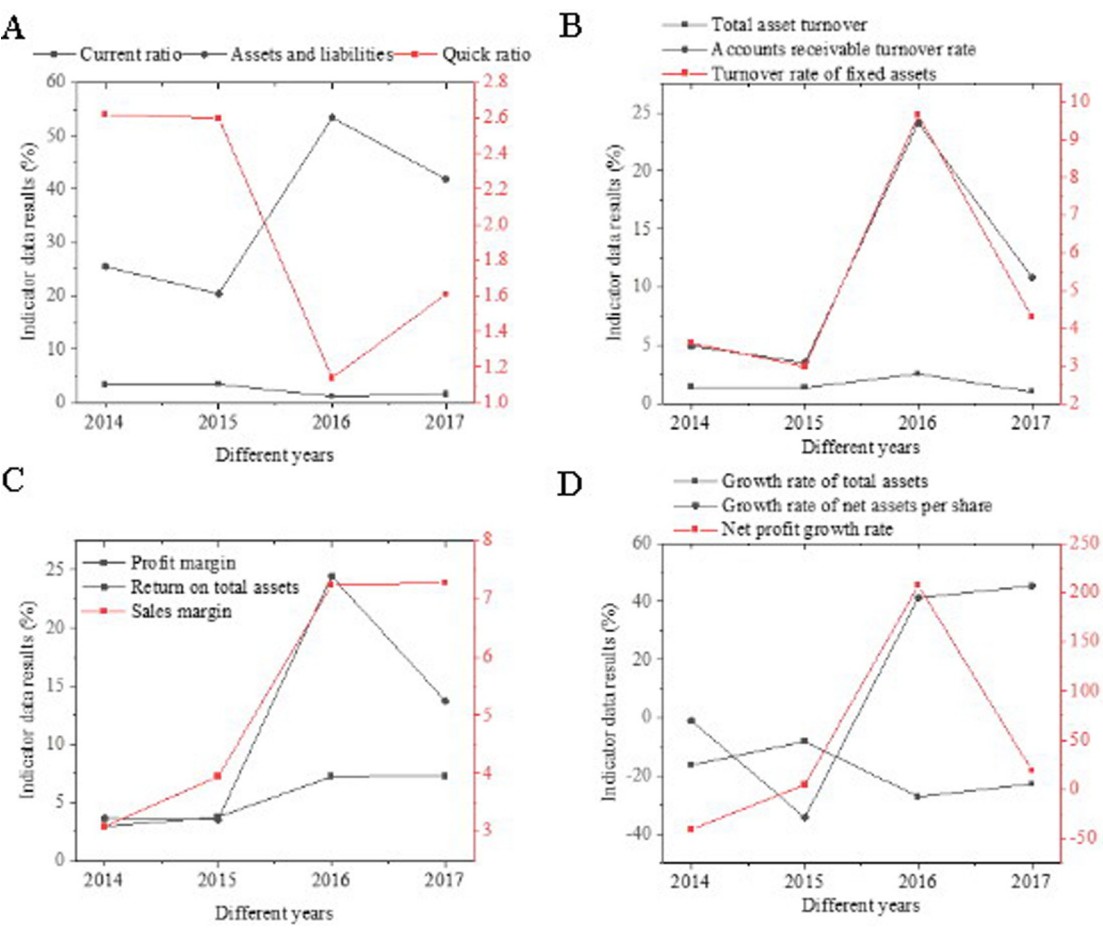

**Fig 5. Analysis results of SF's financial indicator.**

profit margin parallel to its peers, other indicators are significantly behind its peers. The reason is that other small enterprises' competition optimizes its internal structure; thus, the cost has dropped significantly. In terms of growth ability, SF has invested the considerable section resources it borrowed into profit-associated assets, and the scale of assets, profits, and shareholder rights have all made significant progress.

As shown in Fig 7, the non-financial indicators are analyzed as well, where Fig 7A–7C show the satisfaction of large-scale enterprises, the satisfaction of small-scale enterprises, and industry complaints, respectively. It is not difficult to see that among large enterprises, SF has the highest satisfaction, reaching 83.4 points, and its monthly average complaint rate is the lowest, which is far lower than the industry average. STO has the lowest satisfaction, and its monthly average complaint rate is the highest in the industry. Among small enterprises, YUNDA has the highest satisfaction, while TTK has the lowest satisfaction. Thus, SF Express, as a service industry, has well-practiced the mission of customers as the vision of the enterprise.

Fig 8A shows the principal profit indicators of UPS and FedEx and the net profit margin of SF, Fig 8B presents the comparison result of the return on net assets, and Fig 8C exhibits the comparison result of total asset turnover rate. SF is close to UPS and FedEx, the two express delivery giants, in terms of return on net assets and net sales margin; the difference is not as big as that in asset size. This result shows that under the same heavy asset structure, SF's profit indicators are completely world-class. UPS and FedEx's profit indicators are even far behind

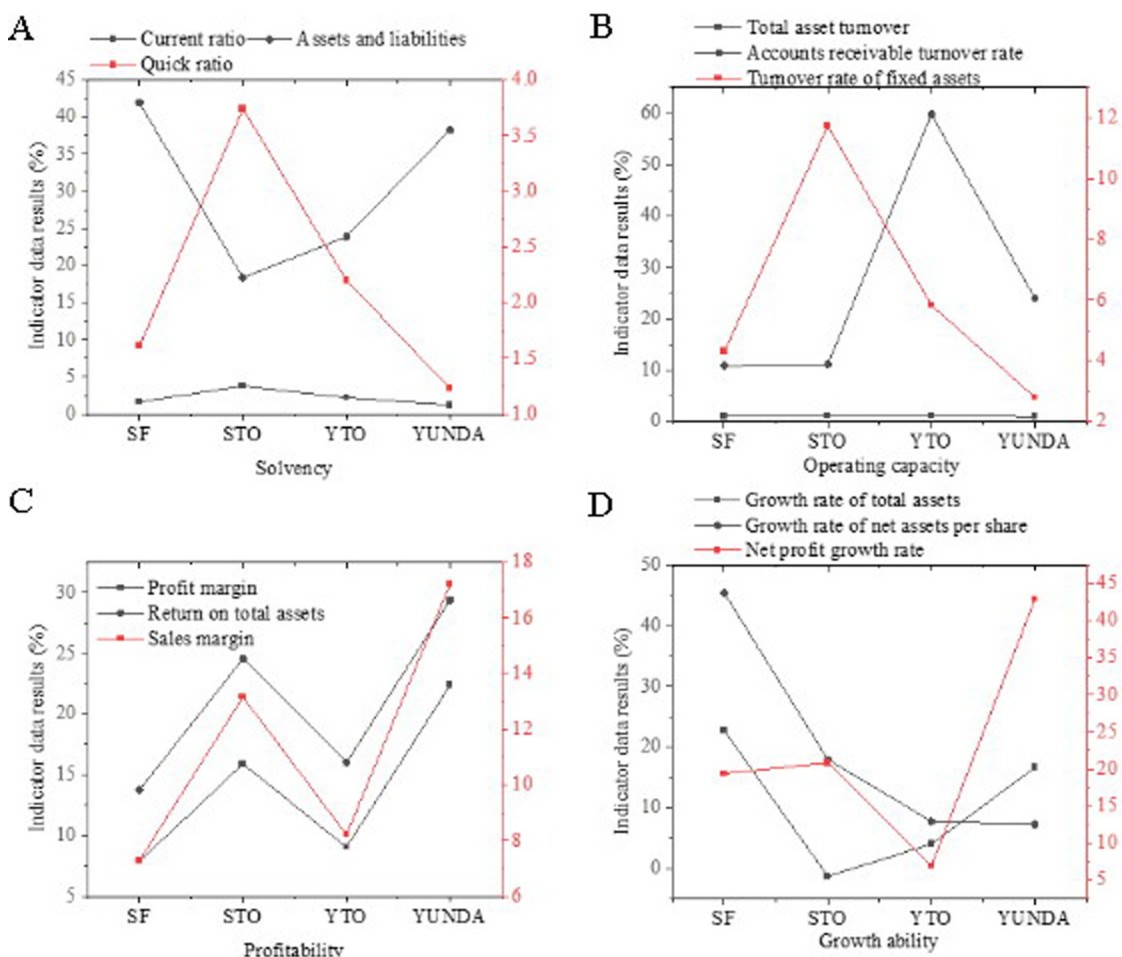

**Fig 6. Comparison of horizontal capabilities of Chinese express enterprises in terms of financial indicators.**

those of STO, YTO, ZTO, and YUNDA due to the differences in business methods; nevertheless, this result does not represent its competitiveness. Thus, under the heavy asset structure, SF has comparatively good asset management capabilities.

As shown in Fig 9, cloud computing is introduced to improve SF enterprise's data processing capabilities further. Fig 9A shows the model performance analysis result, and Fig 9B displays the model data processing effect. Compared with the models proposed in [31–33], cloud computing has effectively improved the express delivery industry's local service data processing capabilities. Besides, the performance of the model has been significantly improved. Compared with the significant data processing method, the model performance is increased by 20%. In terms of local data processing capacity, the time consumed has been shortened from the original 30min to the current10min, an efficiency increase of 33%. This result also shows that the proposed data processing method is more effective.

## Result discussions

In the data processing system, edge computing is a distributed open platform closest to things or data source in the network which integrates the core capabilities of network, calculation, storage, and application, and provides edge intelligent services nearby. The most prominent

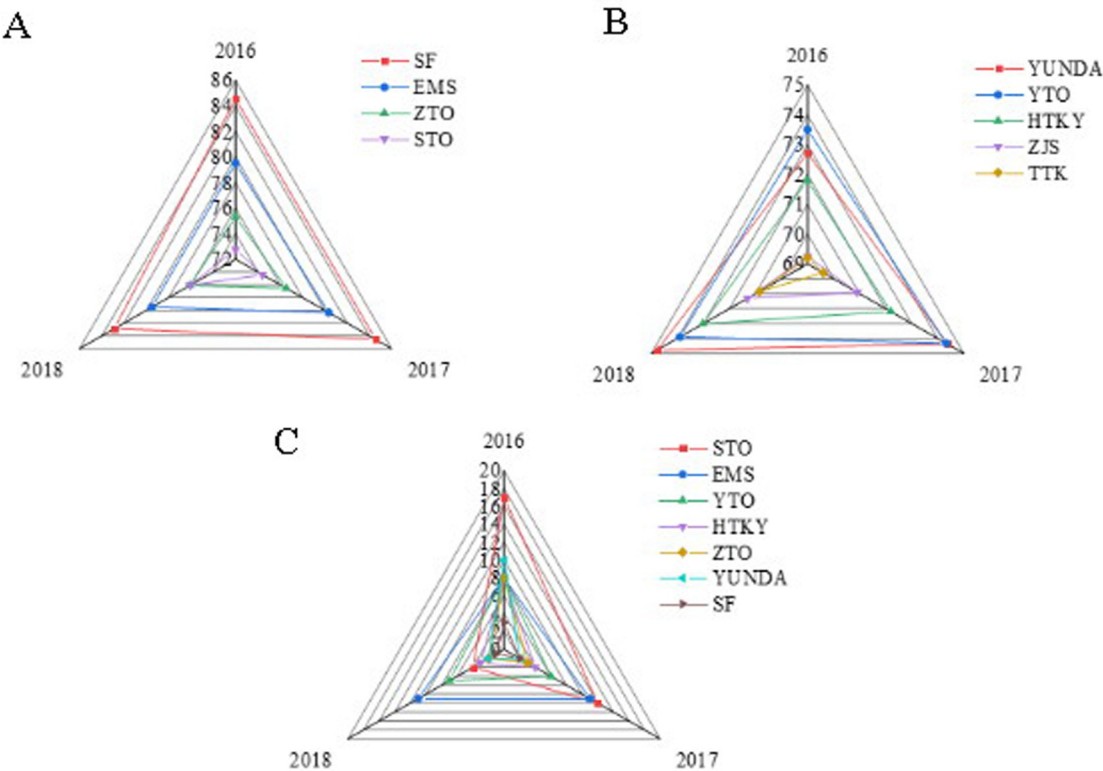

**Fig 7. Comparison of horizontal capabilities of Chinese express enterprises in terms of non-financial indicators.**

feature of edge computing is to provide services on the edge of the network closer to the terminal, which can meet the key needs of various industries in digital agile connection, real-time business, data optimization, application intelligence, and security privacy protection. Its advantages of connecting the physical world and digital world promote intelligence progress. The data processing platform of edge computing enterprises has better performance in processing time and efficiency than the existing models, which are mentioned in the relevant works [34]. Any functional entity between the data source and the cloud computing center can become the edge side of the network. Entities provide real-time, dynamic, and intelligent service computing for terminal customers by carrying edge computing platform with core capabilities of network, computing, storage, and application. The data processing by the devices according to the proximal distribution principle provides a better structure to support data security and privacy protection. The main architecture of the edge computing model mainly includes core infrastructure, edge data center, edge network, and edge devices. This has also been reported in relevant literature. Among them, Qi et al. (2019) believed that different constraints such as bandwidth overfull and delay time limit the availability of cloud-based smart manufacturing service in high-speed and low-delay real-time applications. With the help of cloud computing, fog computing and edge computing, they introduced a hierarchical intelligent reference architecture, which effectively solved the problem of enterprise data processing [35].

In terms of corporate asset relations, asset structure refers to the composition and proportion of various assets. Common indicators reflecting asset structure are the ratio of current assets to total assets, the ratio of long-term assets to total assets, the internal structure of current assets, and the internal structure of long-term assets. The reasonable asset structure can

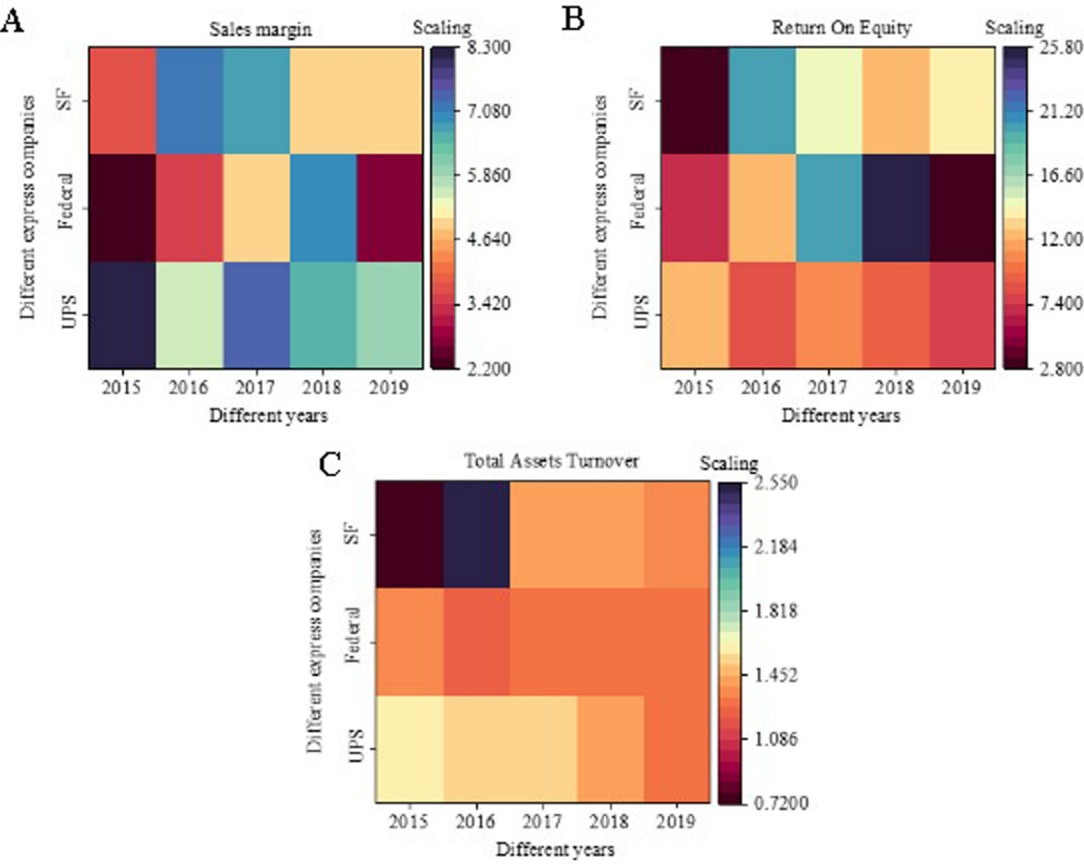

**Fig 8. Financial indicator analysis results of global express enterprises.**

make the assets fully play their role and create more profits for enterprises. In contrast, the unreasonable asset structure not only affects the profitability of enterprises, but also may affect

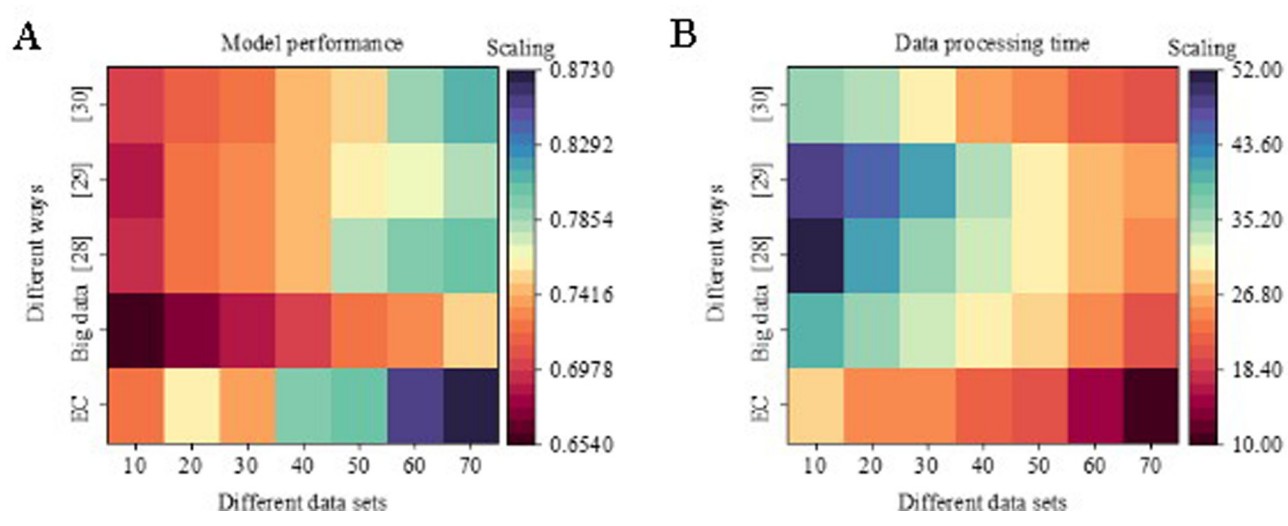

**Fig 9. Analysis results of the cloud computing processing power.**

the normal operation of enterprises, as mentioned in the relevant reports [36]. Excessive current assets can cause idle assets and affect profitability. Fixed assets are necessary to maintain the normal production and operation of enterprises, the main driving force for the creation of enterprise value, and the material basis for the full play of current assets. Therefore, fixed assets have a significant impact on the long-term profitability of enterprises. However, fixed assets generally have relatively poor liquidity, so the high proportion of fixed assets will increase the financial risk of enterprises. Intangible assets refer to identifiable non-monetary assets without physical form owned or controlled by enterprises. In the fierce market competition of brand war and R&D capability war, the moderate increase in the proportion of intangible assets is helpful to improve the competitive strength of enterprises. Since corporate profits mainly depend on tangible assets, it is necessary to control the proportion of intangible assets carefully. However, the improvement of a company's scale, grade, and popularity are inseparable from the impact of patent and trademark rights which will undoubtedly improve the profitability of enterprises, as reported in the relevant works [37]. The following are suggestions for express enterprises based on the research.

Analysis of SF concludes that the rational allocation of asset structure is inseparable from the influence of the enterprise's strategy. The improvement of sustainable profitability is also closely linked to asset structure allocation. An enterprise's strategic choice is an essential basis for it to decide which asset structure to choose. If the enterprise wants to take a cost-leading strategy, it will find ways to reduce costs; the costs are controlled by reducing necessary expenses and meeting the most basic needs. If an enterprise wants to pursue a differentiation strategy, it must shape its brand image and provide high-end differentiated services by improving service quality, increasing research and development investment, improving product design, and ensuring after-sales services. Asset structure is a vital in determining sustainable profitability. As a resource owned and controlled by an enterprise, assets are the foundation for products and services provided by an enterprise. Having a core resource unique in the industry that is difficult to replicate is the greatest guarantee for the sustainable profitability of an enterprise. Sustainable profitability, as a result, will also correctively affect the enterprise's strategy and asset structure. If an enterprise buys huge assets for a strategy but its sustainable profitability does not improve in the end, the enterprise may have made a mistake in its judgment of the industry, or a problem may occur with the utilization rate of assets. At this time, the enterprise needs to reflect on its strategy and asset structure allocation. There is an interaction mechanism among enterprise strategy, asset structure, and sustainable profitability. This effect is not one-way. The enterprise must allocate a reasonable asset structure under the guidance of strategic goals to obtain sustainable profitability. Once the improvement of sustainable profitability falls short of expectations, enterprises must reconsider whether their strategies and the allocation of asset structure are reasonable.

Implications for the express delivery industry in China are put forward. First, enterprises can increase capital investment and transform themselves into asset-heavy models. SF Express serves as a model of the future transformation of other enterprises. With the slowdown of business growth, express delivery enterprises urgently need to find new growth points, such as heavy goods, cold chain, and other new businesses. However, these new businesses often require the support of heavy assets. At present, both ZTO and YUNDA have strengthened their construction of transshipment centers and basically achieved the self-operation of transshipment centers. Moreover, YTO has also self-operated franchisees in some important areas. Regarding customer reviews, the satisfaction of ZTO and YUNDA has indeed improved; still, there is a big gap between them and SF Express. Second, enterprises can increase investment in technology and transform themselves into information platforms. Capitalization is not the only option for express delivery enterprises that adopt asset-light operations. If their information mining

capabilities and service awareness can be improved, the asset-light operation model will still make a big difference. Taking Cainiao, an express service platform, as an example, although it does not have its transportation system, it can provide various express enterprises with accurate deployment information due to the strong data support of Alibaba Group. In the future, if express delivery enterprises can collect statistics on demand for goods, they can understand the demand and circulation of products in a particular area, which is of great value whether to improve logistics routes or provide to third parties. Finally, enterprises can improve the management of franchisees. SF Express is the only enterprise in the industry that has successfully transitioned from a franchise system to a direct management system. It has mature management experience in franchisee management. At present, various asset-light enterprises can learn from SF Express's self-operated model in improving franchisee management.

## Conclusions

The analysis of the express industry proves that there exist difficulties in the data processing of the current express industry. Therefore, mobile edge computing is introduced to analyze the data of the express delivery industry. Different express delivery enterprises are analyzed from the perspective of strategic management accounting to clarify the impact of different asset allocations on corporate strategy. Besides, the financial data of enterprises is used to explore the sustainable profitability of enterprises. The biggest difference between SF Express and other express companies in China is that its total assets occupy the largest proportion, accounting for more than 45%. This indicates that heavy assets can improve the profitability of enterprises to a certain extent. Through the analysis of financial and non-financial indicators of SF Express, the growth rate of net assets per share of SF Express has maintained at more than 40% in the past two years. The results validate that the performance of the model based on the mobile edge computing has improved by 20%. In terms of local data processing ability, the processing time is decreased by 33% from 30 min to 10 min. The strategic choice of enterprises is the basis for enterprises to decide the asset structure, while the asset structure is an important factor determining sustainable profitability. Furthermore, the strength of sustainable profitability will also have a certain correction effect on corporate strategy and asset structure.

Although some results have been achieved through research based on edge computing from the perspective of strategic management accounting, there are still many deficiencies remaining to be optimized. First, there may be bias and errors in the analysis of the operation mode of heavy assets and light assets, and there may be some aspects out of consideration in the evaluation of the sustainable profitability of enterprises. The main reason is that there is no appropriate data set for unified analysis and management of data. Second, the strategic management accounting theory involves various contents in addition to the six angles discussed in the research, so there lack contents such as market positioning analysis and target management. With the development of edge computing technology and strategic management accounting theory, there will be more scientific strategic analysis tools applied to case studies. Asset structure is a crucial factor in determining sustainable profitability. Assets, as the resources owned and controlled by enterprises, are the foundation for enterprises to produce products and provide services. The core resources that are unique and hard to replicate in the industry are the biggest guarantee for enterprises' sustainable profitability. Sustainable profitability as a result will also have a correction effect on corporate strategy and asset structure. If the huge asset bought by the enterprise cannot improve its sustainable profitability finally, it is possible that the enterprise has judgment on the industry, or there is a problem in the utilization rate of assets. Correspondingly, the enterprise needs to reflect on its own strategy and

asset structure. Future research can focus on the comprehensive and scientific evaluation of the sustainable profitability of enterprises, and explore the increasingly clear competition pattern of express delivery industry, which will deepen scholars' understanding of the differences in asset structure.

## Supporting information

**S1 Data.**
(RAR)

## Author Contributions

**Conceptualization:** Wenquan Shi.

**Data curation:** Wenquan Shi.

**Formal analysis:** Wenquan Shi.

**Methodology:** Wenquan Shi.

**Project administration:** Wenquan Shi.

**Resources:** Wenquan Shi.

**Software:** Wenquan Shi.

**Supervision:** Wenquan Shi.

**Validation:** Wenquan Shi.

**Visualization:** Wenquan Shi.

**Writing – original draft:** Wenquan Shi.

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
