## [Decision Letter · Decision Letter 0]

14 Jun 2021

PONE-D-21-06810

Analyzing Enterprise Asset Structure and Profitability Using Cloud Computing and Strategic Management Accounting

PLOS ONE

Dear Dr. Shi,

Thank you for submitting your manuscript to PLOS ONE. After careful consideration,I feel that it has merit but does not fully meet PLOS ONE’s publication criteria as it currently stands.

Both reviewers consider that there is a lack of motivation in this research. I cannot guarantee that after the revision the paper will be suitable of publication considering the concerns showed by one of the reviewers regarding to the methodology used, specifically, with the dada considered.

Authors must provide a reasonable explanation of these major concerns.

Other minor questions are that paper should include more discussion engaging in the dialogue with the relevant literature as well as the future perspectives and limitations of this research. Therefore, I invite you to submit a revised version of the manuscript that addresses the points raised during the review process.

We look forward to receiving your revised manuscript.

Kind regards,

J E. Trinidad Segovia

Academic Editor

PLOS ONE

Journal Requirements:

Reviewers' comments:

Reviewer's Responses to Questions

**Comments to the Author**

1. Is the manuscript technically sound, and do the data support the conclusions?

Reviewer #1: Yes

Reviewer #2: Partly

2. Has the statistical analysis been performed appropriately and rigorously? 

Reviewer #1: Yes

Reviewer #2: No

3. Have the authors made all data underlying the findings in their manuscript fully available?

Reviewer #1: Yes

Reviewer #2: No

4. Is the manuscript presented in an intelligible fashion and written in standard English?

Reviewer #1: Yes

Reviewer #2: No

5. Review Comments to the Author

Reviewer #1: This research is mainly for the financial analysis of express delivery companies, mainly explores the financial indicators and non-financial indicators of express delivery companies, and analyzes the factors that affect the sustainable profitability of the express delivery industry. At the same time, the article uses the current advanced cloud computing data processing methods to analyze the market competitiveness and commercial value of express delivery companies, and conducts a more comprehensive and scientific analysis for the express delivery companies studied. I think the research and analysis method of this article is reasonable and scientific, but I think this article can be published after modification.

The main problems of this article are as follows:

1. It is recommended that authors include specific values in the abstract, as this can improve the scientific nature of the article. It is recommended that the authors modify this.

2. In the introduction, the author stated that "In 2019, express service enterprises’ business volume reached 50.71 billion pieces, with a year-on-year increase of 26.6%." The author needs to explain the source of this data.

3. The last paragraph in the introduction does not need to explain all the research content, only the main purpose and method of the research need to be explained. I suggest the author to delete this part.

4. In terms of literature review, the author needs to explain the previous research results, and the limitations of the previous research and the improvements that need to be made.

5. The enterprise analysis model is the key content of the research, and the author needs to explain the model in Figure 1 more completely.

6. The author needs to explain the specific parameters and steps of cloud computing data processing used in the research.

7. It would be more appropriate to change the "Strategic management accounting analysis" section of the article to "research data sources".

8. What is the experimental environment for cloud computing processing used? please explain carefully.

9. The conclusion is an explanation of the research results of the article, so the author needs to explain the specific results of the research in the conclusion.

Reviewer #2: 1. The authors had better rewrite the abstract section to appeal to readers. In other words, authors have to explain why your findings matter as well as why your work would contribute to the existing literature.

2. Readers prefer finding the strong motivation, purpose, importance of this study, and even the contribution of this study in the introduction section.

3. Selecting total assets, current assets, and intangible assets to analyze the enterprise asset structure might be reasonable; however, accounts payable might not be appropriately selected to analyze asset structure due to account payable is not an asset item in the balance sheet.

4. Why do authors select debt repayment, operation, profitability, and growth capabilities to analyze enterprise profitability? Authors have to provide supporting arguments for these items except profitability.

5. Reviewers prefer statistical results with enough samples instead of trend charts, pie charts, etc. since the latter (i.e., trend charts, pie charts, etc.) would be difficult to persuade readers. In addition, Figures 1-11 are not shown in the charts shown at the bottom of this paper, which might not be welcomed by reviewers.

6. Providing and comparing descriptive statistics of relevant asset items for different years and different companies as your revealed findings might not be welcomed by academic journals.

7. In addition to implication sections, this paper should include more discussion engaging in the dialogue with the relevant literature.

8. Proposing future perspectives for the study should be mentioned specifically; in addition, authors have to tell readers why such future perspectives are worthwhile for further research as well.

6. PLOS authors have the option to publish the peer review history of their article (what does this mean?). If published, this will include your full peer review and any attached files.

Reviewer #1: **Yes: **Xin Gao

Reviewer #2: No

---

## [Author Response · Author response to Decision Letter 0]

29 Jul 2021

Reviewer #1: This research is mainly for the financial analysis of express delivery companies, mainly explores the financial indicators and non-financial indicators of express delivery companies, and analyzes the factors that affect the sustainable profitability of the express delivery industry. At the same time, the article uses the current advanced cloud computing data processing methods to analyze the market competitiveness and commercial value of express delivery companies, and conducts a more comprehensive and scientific analysis for the express delivery companies studied. I think the research and analysis method of this article is reasonable and scientific, but I think this article can be published after modification.

The main problems of this article are as follows:

1. It is recommended that authors include specific values in the abstract, as this can improve the scientific nature of the article. It is recommended that the authors modify this.

Response: thank you for your valuable advice! We have included certain values in the abstract to demonstrate the scientific nature of our work. 

2. In the introduction, the author stated that "In 2019, express service enterprises’ business volume reached 50.71 billion pieces, with a year-on-year increase of 26.6%." The author needs to explain the source of this data.

Response: thank you for your valuable suggestions! We have added references to the relevant data in the introduction to illustrate this phenomenon. 

3. The last paragraph in the introduction does not need to explain all the research content, only the main purpose and method of the research need to be explained. I suggest the author to delete this part.

Response: thank you for your valuable opinion! We have supplemented the description of the main objectives and methods of the present study in the last paragraph of the introduction as required. 

4. In terms of literature review, the author needs to explain the previous research results, and the limitations of the previous research and the improvements that need to be made.

Response: thank you for your valuable advice! We have added the limitations of the current research and the areas for improvement in the literature review “A Summary of Previous Works”, according to your suggestions.

5. The enterprise analysis model is the key content of the research, and the author needs to explain the model in Figure 1 more completely.

Response: thank you for your valuable advice! We have supplemented the detail explanation of the current model in Figure 1 as suggested. 

6. The author needs to explain the specific parameters and steps of cloud computing data processing used in the research.

Response: thank you for your valuable advice! We have added the specific parameters and corresponding steps for cloud computing data processing. 

7. It would be more appropriate to change the "Strategic management accounting analysis" section of the article to "research data sources".

Response: thank you for your valuable comment! We have changed the strategic management accounting analysis to research data sources as required. 

8. What is the experimental environment for cloud computing processing used? please explain carefully.

Response: thank you for your valuable opinions! We have supplemented the detail description of the experimental environment for cloud computing processing. 

9. The conclusion is an explanation of the research results of the article, so the author needs to explain the specific results of the research in the conclusion.

Response: thank you for your valuable advice! We have included specific numerical values in our conclusions to demonstrate the reliability of our research. 

Reviewer #2: 1. The authors had better rewrite the abstract section to appeal to readers. In other words, authors have to explain why your findings matter as well as why your work would contribute to the existing literature.

Response: thank you for your valuable suggestions! We have revised the abstract of the paper and illustrated the key findings of the current study and the major contributions to the current field. 

2. Readers prefer finding the strong motivation, purpose, importance of this study, and even the contribution of this study in the introduction section.

Response: thank you for your valuable opinions! We have re-stated the main motivation, purpose and innovation of this research in the introduction. 

3. Selecting total assets, current assets, and intangible assets to analyze the enterprise asset structure might be reasonable; however, accounts payable might not be appropriately selected to analyze asset structure due to account payable is not an asset item in the balance sheet.

Response: thank you for your valuable advice! Considering that accounts payable is also a key index in financial analysis, which determines the liquidity performance of the company's assets, so we finally choose this index for analysis. 

4. Why do authors select debt repayment, operation, profitability, and growth capabilities to analyze enterprise profitability? Authors have to provide supporting arguments for these items except profitability.

Response: thank you for your valuable suggestion! We have added relevant references in the method section to illustrate that the selection of relevant indicators for the evaluation of corporate profitability is scientific. 

5. Reviewers prefer statistical results with enough samples instead of trend charts, pie charts, etc. since the latter (i.e., trend charts, pie charts, etc.) would be difficult to persuade readers. In addition, Figures 1-11 are not shown in the charts shown at the bottom of this paper, which might not be welcomed by reviewers.

Response: thank you for your valuable comments! The Figure 1-11 was provided to the Editorial Department in the form of attachment, which is the journal requirement. We have added the corresponding chart content in the article, and diversified our statistical results displayed in various forms.

6. Providing and comparing descriptive statistics of relevant asset items for different years and different companies as your revealed findings might not be welcomed by academic journals.

Response: thank you for your valuable advice! We have excluded the descriptive statistical results of different company asset items in different years. 

7. In addition to implication sections, this paper should include more discussion engaging in the dialogue with the relevant literature.

Response: thank you for your valuable opinions! We have included the content of discussion and compared the results of this paper with previous studies to further verify the effectiveness of the model proposed in this paper. 

8. Proposing future perspectives for the study should be mentioned specifically; in addition, authors have to tell readers why such future perspectives are worthwhile for further research as well.

Response: thank you for your valuable suggestions! We have rewritten the conclusion of the paper and detailed the need for future research.

---

## [Decision Letter · Decision Letter 1]

23 Aug 2021

PONE-D-21-06810R1

Analyzing Enterprise Asset Structure and Profitability Using Cloud Computing and Strategic Management Accounting

PLOS ONE

Dear Dr. Shi,

Thank you for submitting your manuscript to PLOS ONE. After careful consideration, I feel that it has merit but does not fully meet PLOS ONE’s publication criteria as it currently stands.

Most of the reviewers’ comments have been addressed. However, remains a few minor questions that need the attention of the authors. Therefore, I invite you to submit a revised version of the manuscript that addresses the points raised during the review process.

We look forward to receiving your revised manuscript.

Kind regards,

J E. Trinidad Segovia

Academic Editor

PLOS ONE

Journal Requirements:

Additional Editor Comments (if provided):

Reviewers' comments:

Reviewer's Responses to Questions

**Comments to the Author**

1. If the authors have adequately addressed your comments raised in a previous round of review and you feel that this manuscript is now acceptable for publication, you may indicate that here to bypass the “Comments to the Author” section, enter your conflict of interest statement in the “Confidential to Editor” section, and submit your "Accept" recommendation.

Reviewer #1: (No Response)

Reviewer #2: (No Response)

2. Is the manuscript technically sound, and do the data support the conclusions?

Reviewer #1: Yes

Reviewer #2: Yes

3. Has the statistical analysis been performed appropriately and rigorously? 

Reviewer #1: Yes

Reviewer #2: N/A

4. Have the authors made all data underlying the findings in their manuscript fully available?

Reviewer #1: Yes

Reviewer #2: Yes

5. Is the manuscript presented in an intelligible fashion and written in standard English?

Reviewer #1: Yes

Reviewer #2: Yes

6. Review Comments to the Author

Reviewer #1: In this revision, authors have already addressed all the comments. The paper can be accepted now.The study presents the results of original research.

Reviewer #2: Comments

1. The author should shorten your abstract and highlight your concern in this section since the abstract with many contexts might not easily appeal to readers. For example, the author might explain the revealed results with supporting arguments instead of listing statistical numbers (i.e., 51.8%, 45%, etc.) in detail for each finding.

2. We suggest that you had better tell readers the outlay of this paper at the end of the introduction section.

3. According to the format of PLoS One, the authors should not cite references as presented below (please lines 103-115) , and readers can not find these citations in the Reference section.

4. The literature section should include the relevant studies rather than current studies only. We then suggest that the subtitle of this section might not name “Current research…..”, but we agree that the author could propose current research progress in the context of this section.

5. It might not be appropriate that there is only one long paragraph in the Conclusion section. In addition, the author should use “First, Second……” instead of “First, Secondly……” in this section.

7. PLOS authors have the option to publish the peer review history of their article (what does this mean?). If published, this will include your full peer review and any attached files.

Reviewer #1: No

Reviewer #2: No

---

## [Author Response · Author response to Decision Letter 1]

26 Aug 2021

Reviewer #2: Comments

1. The author should shorten your abstract and highlight your concern in this section since the abstract with many contexts might not easily appeal to readers. For example, the author might explain the revealed results with supporting arguments instead of listing statistical numbers (i.e., 51.8%, 45%, etc.) in detail for each finding.

Reply: the "abstract" part has been adjusted as required.

2. We suggest that you had better tell readers the outlay of this paper at the end of the introduction section.

Reply: a description of the outlay of this paper has been added at the end of the "Introduction".

3. According to the format of PLoS One, the authors should not cite references as presented below (please lines 103-115) , and readers can not find these citations in the Reference section.

Reply: (original lines 103-115) the reference was replaced as required.

4. The literature section should include the relevant studies rather than current studies only. We then suggest that the subtitle of this section might not name “Current research…..”, but we agree that the author could propose current research progress in the context of this section.

Reply: the subtitle of literature section has been modified as required and the relevant literature is supplemented.

5. It might not be appropriate that there is only one long paragraph in the Conclusion section. In addition, the author should use “First, Second……” instead of “First, Secondly……” in this section.

Reply: the conclusion section has been segmented as required, and the description of "First, Secondly..." has been modified to "First, Second...".

---

## [Decision Letter · Decision Letter 2]

13 Sep 2021

Analyzing Enterprise Asset Structure and Profitability Using Cloud Computing and Strategic Management Accounting

PONE-D-21-06810R2

Dear Dr. Shi,

We’re pleased to inform you that your manuscript has been judged scientifically suitable for publication and will be formally accepted for publication once it meets all outstanding technical requirements.

Kind regards,

J E. Trinidad Segovia

Section Editor

PLOS ONE

Additional Editor Comments (optional):

Reviewers' comments:

Reviewer's Responses to Questions

**Comments to the Author**

1. If the authors have adequately addressed your comments raised in a previous round of review and you feel that this manuscript is now acceptable for publication, you may indicate that here to bypass the “Comments to the Author” section, enter your conflict of interest statement in the “Confidential to Editor” section, and submit your "Accept" recommendation.

Reviewer #2: All comments have been addressed

2. Is the manuscript technically sound, and do the data support the conclusions?

Reviewer #2: Yes

3. Has the statistical analysis been performed appropriately and rigorously? 

Reviewer #2: Yes

4. Have the authors made all data underlying the findings in their manuscript fully available?

Reviewer #2: Yes

5. Is the manuscript presented in an intelligible fashion and written in standard English?

Reviewer #2: Yes

6. Review Comments to the Author

Reviewer #2: Your revised version looks OK. However, we suggest that the outlay of this paper (e.g., Section 2 surveys the relevant studies. Data and methodologies are introduced in Section 3.........) had better presented at the end of Section 1.

In addition, the presentation of your table in the context is not professional since only horizontal lines without vertical lines would be shown in your tables. If you refer to other academic journals indexed SSCI/SCI, your table presentation is seldom shown in these journals. Likewise, the outlay of this paper is rarely undisclosed at the end of section 1.

7. PLOS authors have the option to publish the peer review history of their article (what does this mean?). If published, this will include your full peer review and any attached files.

Reviewer #2: No

---

## [Editor Report · Acceptance letter]

22 Sep 2021

PONE-D-21-06810R2 

Analyzing Enterprise Asset Structure and Profitability Using Cloud Computing and Strategic Management Accounting 

Dear Dr. Shi:

I'm pleased to inform you that your manuscript has been deemed suitable for publication in PLOS ONE. Congratulations! Your manuscript is now with our production department. 

Kind regards, 

on behalf of

Dr. J E. Trinidad Segovia 

Section Editor

PLOS ONE